# *TIE1* as a Candidate Gene for Lymphatic Malformations with or without Lymphedema

**DOI:** 10.3390/ijms21186780

**Published:** 2020-09-16

**Authors:** Sandro Michelini, Maurizio Ricci, Dominika Veselenyiova, Sercan Kenanoglu, Danjela Kurti, Mirko Baglivo, Alessandro Fiorentino, Syed Hussain Basha, Sasi Priya, Roberta Serrani, Juraj Krajcovic, Munis Dundar, Astrit Dautaj, Matteo Bertelli

**Affiliations:** 1Department of Vascular Rehabilitation, San Giovanni Battista Hospital, 00148 Rome, Italy; s.michelini@acismom.it (S.M.); a.fiorentino@acismom.it (A.F.); 2Division of Rehabilitation Medicine, Azienda Ospedaliero-Universitaria, Ospedali Riuniti di Ancona, 60126 Ancona, Italy; maurizio.ricci@ospedaliriuniti.marche.it (M.R.); roberta.serrani@ospedaliriuniti.marche.it (R.S.); 3Department of Biology, Faculty of Natural Sciences, University of Ss. Cyril and Methodius In Trnava, 91701 Trnava, Slovakia; d.veselenyiova@gmail.com (D.V.); juraj.krajcovic@ucm.sk (J.K.); 4MAGI Euregio, 39100 Bolzano, BZ, Italy; sercankenan@gmail.com (S.K.); genetica.clinica@assomagi.org (D.K.); mirko.baglivo@assomagi.org (M.B.); matteo.bertelli@assomagi.org (M.B.); 5Department of Medical Genetics, Faculty of Medicine, Erciyes University, 38039 Kayseri, Turkey; dundar@erciyes.edu.tr; 6MAGI-Balkan, 1019 Tirana, Albania; 7Innovative Informatica Technologies, Hyderabad 500 049, India; shb@innovativeinformatica.com (S.H.B.); sasipriya641997@gmail.com (S.P.); 8EBTNA-Lab, 38068 Rovereto, TN, Italy; 9MAGI’s Lab, 38068 Rovereto, TN, Italy

**Keywords:** TIE1, NGS, lymphedema, genetic diagnostics

## Abstract

TIE1 is a cell surface protein expressed in endothelial cells. Involved in angiogenesis and lymphangiogenesis, including morphogenesis of lymphatic valves, TIE1 is important for lymphatic system functional integrity. The main purpose of this study was to identify different variants in the *TIE1* gene that could be associated with lymphatic malformations or dysfunction and predisposition for lymphedema. In a cohort of 235 Italian lymphedema patients, who tested negative for variants in known lymphedema genes, we performed a further test for new candidate genes, including *TIE1*. Three probands carried different variants in *TIE1*. Two of these segregated with lymphedema or lymphatic dysfunction in familial cases. Variants in *TIE1* could contribute to the onset of lymphedema. On the basis of our findings, we propose *TIE1* as a candidate gene for comprehensive genetic testing of lymphedema.

## 1. Introduction

The *TIE1* gene, located on the short arm of chromosome 1 (1p34.2), encodes a tyrosine kinase receptor with an immunoglobulin and EGF factor homology domain. The TIE1 protein occurs on the surface of endothelial cells and was cloned for the first time in 1992 by Partanen and colleagues [1]. *TIE1* is predominantly expressed in endothelial cells of blood and lymphatic vessels and in several lines of hematopoietic cells.

TIE1 is important for angiogenesis and lymphangiogenesis, because together with TIE2 and angiopoietins (ANGs), it takes part in the ANG-TIE signaling pathway. Interestingly, no ligand that binds TIE1 has yet been identified, making TIE1 an orphan receptor [2]. However, TIE1 interacts directly with another member of the tyrosine kinase receptor family, TIE2. TIE2 binds various angiopoietins, including Ang1 and Ang2. ANGs can activate the formation of the TIE signaling complex in the junctions of endothelial cells. The TIE complex in turn regulates the survival signal of endothelial cells [3,4]. The normal function of TIE1 is therefore directly and indirectly important for normal vascular development, since it affects the function of TIE2 and angiopoietins [2]. The ANG-TIE signaling pathway is involved in homeostasis, inflammatory response, angiogenesis and lymphangiogenesis; abnormal ANG-TIE signaling can give rise to pathological processes [2]. 

More insight into the biological functions of TIE1 has come from mouse models (Table 1). First, the role of *TIE1* in the vascular system was demonstrated. *TIE1^−/−^* mice showed severe edema and bleeding and died during gestation (E13.5). Death was due to hemorrhage and abnormal formation of microvessels [5,6]. The role of *TIE1* in the lymphatic system was subsequently demonstrated. D’Amico et al. showed that deletion of *TIE1* in mice has a major impact on the formation of lymphatic vasculature. *TIE1^−/−^* embryos developed edema at E12.5 in the dorsal body and neck area. The authors reported that *TIE1* was expressed in lymphatic sacs during embryo development and that the lymphatic sacs showed impaired patterning in *TIE1*-null mice. TIE1-deficiency also led to impaired development and integrity of the lymphatic capillaries. A similar phenotype was reported in animals with conditional *TIE1* mutation [7]. These findings demonstrate that *TIE1* is indispensable for lymphangiogenesis during embryogenesis. 

In another study, Qu et al. generated hypomorphic mice with low *TIE1* expression, as well as conditional *TIE1* mutant mice. Insertion of a neo cassette (*TIE1^neo/neo^*) caused abnormal splicing and reduction of *TIE1* expression to ~20% of normal. This resulted in abnormalities in the lymphatic vasculature and overgrown jugular lymphatic vessels. Furthermore, insufficient expression of *TIE1* led to failure of fluid drainage in the skin of mutant mice, causing edema. Interestingly, an abnormal cardiovascular phenotype was not observed in mice with reduced *TIE1* expression [8]. 

These results clearly demonstrate the importance of *TIE1* in the normal development of the lymphatic system, especially the lymphatic vasculature. Recent studies also showed that *TIE1* is indispensable for the morphogenesis of lymphatic valves [9]. It was reported that while *TIE1* is expressed ubiquitously in early embryonic development of the lymphatic system, there is a significant increase in *TIE1* expression in lymphatic endothelial cells at the onset of lymphatic valve formation. *TIE1* mutant mice lack lymphatic vessel remodeling capacity, show abnormal lymphatic valve morphogenesis and impaired maturation of the lymphatic system, and fail to develop collecting lymphatic vessels [9,10]. Taken together, *TIE1* mouse models demonstrate that *TIE1* is crucial for lymphatic system formation in a dose-dependent manner and is especially important for the normal function and formation of lymphatic vessels and valves. 

The network of lymphatic vessels and capillaries is an essential part of the lymphatic system. Lymphatic capillaries are permeable and allow protein-rich interstitial fluid to enter the lymphatic system. This is important for fluid homeostasis and normal immune function. Lymphatic valves are an element of lymphatic-collecting vessels and their role is to maintain retrograde lymph flow and prevent blockage of lymph flow [11]. Lymphatic abnormalities can give rise to diseases such as lymphedema [12]. 

Lymphedema is a lymphatic system disorder caused by accumulation of interstitial fluid. It is a chronic disease with clinical symptoms such as edema, inflammation and fibrosis. The swelling usually affects the extremities and can be accompanied by pain and functional impairment. Fluid accumulates due to lymphatic system malformations such as lymphatic valve abnormalities [13], excessive lymphangiogenesis, or complete or partial absence of lymphatic capillaries and collecting vessels [14].

Several genes are known to be involved in the onset of, or predisposition for, lymphedema. Current genetic testing of lymphedema patients and their family members includes as many as 29 known lymphedema-associated genes [15]. Despite this large number, variants in those genes can usually only explain the etiology of 25–30% patents with lymphedema [16,17]. Therefore, additional genes, as we show, are certainly involved and need to be included to improve test accuracy. 

In our study, we tested 246 Italian lymphedema patients for the known lymphedema genes, but 235 did not carry pathogenic variants. We therefore tested these 235 probands for candidate genes including *TIE1*. Here we demonstrate a link between TIE1 and lymphatic system malformations, indicating the potential of *TIE1* as a candidate gene for genetic testing of lymphedema.

## 2. Results

### 2.1. Clinical Results

We retrospectively enrolled 235 Italian patients diagnosed with lymphedema in our study. These patients had previously been tested for known lymphedema-associated genes and were found negative. We performed a second genetic test for new candidate genes, including *TIE1*. We found *TIE1* variants in three out of 235 probands. The clinical features of probands and tested family members are summarized in Table 2. 

The proband of the first family (female, 23 years) was diagnosed with edema of the lower limbs at age 13 and carried a missense heterozygous *TIE1* variant NM_001253357.1:c.1306C>T. The variant causes a change from arginine to cysteine at position 436 of the protein. According to GnomAD, the frequency of this allele is 0.0000319 and the variant is known in dbSNP as rs139244400. Polyphen and SIFT predict the variant to be deleterious and possibly damaging, respectively. We also tested both parents, the brother and a grandmother of the proband (Figure 1). We found that while the father and brother were negative for *TIE1* variants, the mother and grandmother carried the same variant as the proband. We performed lymphoscintigraphy of the mother, defined as healthy, to investigate her lymphatic system function. The results showed mild deficits of the lymphatic system development, compatible however with good overall clinical compliance. In addition, the grandmother reported episodes of cyclic edema.

The second proband (female, 52 years) has had right lower limb edema since age 25 years. Since this is a sporadic case, no other family members were tested. The proband was found to carry a heterozygous single nucleotide missense variant in *TIE1* (NM_001253357.1:c.3046G>A) that results in a change of glutamine 1016 to lysine. The frequency of this allele is 0.0000489 (GnomAD), and it is listed in dbSNP as rs760492428. SIFT predicts the variant to be deleterious, and according to PolyPhen it is possibly damaging. 

In the third family, we tested the proband and both parents (Figure 2). The proband (female, 47 years) suffers from lymphedema of lower limbs, which started at the knees, diagnosed at age 15. She carries a heterozygous missense variant NM_001253357.1:c.3191G>A that causes a change of arginine 1064 to histidine. The case is familial: the proband’s mother has lymphedema and was found to carry the same *TIE1* variant as the proband. SIFT and PolyPhen predictions characterized this variant as deleterious and probably damaging, respectively. GnomAD reports the frequency of this variant as 0.000772 and its dbSNP ID is rs34993202. All the *TIE1* variants identified in this study are shown in Table 3. 

### 2.2. In-Silico Analysis, Template Selection and Model Building

A template search with BLAST and HHBlits was performed against the SWISS-MODEL template library (SMTL, last update: 24 October 2019, last included PDB release: 18 October 2019). The target sequence (Table 4) was searched against the primary amino acid sequence contained in the SMTL. A total of 13,529 templates matching with different sequence identity and quality percentages were found. Details of the top 10 templates are shown in Table 5.

Based on the percentage of sequence identity, similarity and best quality square, the 1fvr.1.A chain was selected to align the template and query sequences for model building. The final model is shown in Figure 2. Then, we entered the model in Discovery studio visualizer to generate Arg436Cys (Figure 3), Glu1016Lys (Figure 4) and Arg1064His (Figure 5) versions of its structure. Molecular level interaction analysis between native/variant residues was performed (Figure 3, Figure 4 and Figure 5 show snapshots). Details of the residues involved in interactions along with the type of bonds they formed and bond lengths in angstrom units are listed in Table 6, Table 7 and Table 8, respectively.

## 3. Discussion

Lymphedema is a progressive disease that affects approximately 1 in 100,000 individuals. It is characterized by accretion of lymphatic fluid in tissues, causing swelling, inflammation and fibrosis [18]. Despite the many studies into the genetic background of lymphedema, the underlying molecular mechanisms are still unclear. We used Next-Generation Sequencing to determine the genotype of 246 Italian lymphedema patients with regard to known lymphedema genes [15]. Surprisingly, 235 of the patients tested negative for variants in known genes. We therefore decided to investigate new candidate genes, including *TIE1* (OMIM 600222).

The protein TIE1 is a receptor tyrosine kinase, with a role in cardiovascular and lymphatic system development. It is an orphan receptor since no direct binding of ligands by TIE1 has been discovered [1,2]. TIE1 forms a complex with TIE2, another member of the TIE family, which despite its structural similarities with TIE1, directly binds various ligands. Angiopoietins (ANG) have been shown to bind TIE2 and activate the ANG-TIE signaling pathway [19]. ANG-TIE signaling is important for lymphatic vessel network development, especially for the remodeling and maintenance of lymphatic collecting vessels [20].

Development of the mammalian lymphatic system is a highly conserved process and the importance of *TIE1* function has been demonstrated by mouse models in different situations [7,8,9,10,21]. All in all, since animal models of *TIE1* clearly show its function in the morphogenesis, remodeling and maturation of lymphatic vessels, *TIE1* has been suggested as a gene involved in the development of lymphedema [9].

In our study, we identified three different heterozygous missense *TIE1* variants in 235 lymphedema patients (3/235; 1.28%), who tested negative for known lymphedema genes. All three variants caused a change in amino acids in the resulting protein. The probands were all female and diagnosed with lymphedema of one or both lower limbs at an early age (before 25 years). 

In the first family we also found the same rs139244400 variant carried by the proband in the mother and grandmother, who do not have overt lymphedema, although lymphoscintigraphy of the mother confirmed mild deficits of the lymphatic system and the grandmother reported episodes of lymphatic impairment.

For the second proband, a sporadic case carrying the variant rs760492428, no family members were tested.

In the third family, the case is familial as the mother of the proband also suffers from lymphedema. The rs34993202 variant appears to segregate with lymphedema in this family. 

To further evaluate the effects of the variants on the overall structure and function of the resulting protein, we performed comprehensive bioinformatic analysis. Our in-silico analysis showed that the *TIE1* gene codes a structure in which Arg436 is much more stable than the variant Cys436: the gene coded with Arg436 shows 11 interactions whereas the variant shows only five, and all except two interactions, i.e. those with Arg388 and Glu425, are quite different from each other and even the two similar interactions show a slight difference in bond length. The same was observed in the case of Glu1016Lys, where Glu1016 has 11 interactions, whereas Lys1016 only shows nine, but compared to the Arg436Cys variant, Lys1016 in the Glu1016Cys variant shows six similar interactions, like Glu1016, but varying in bond length. In case of the Arg1064His variant, Arg1064 shows five interactions whereas His1064 shows only three, two quite similar to Arg1064, i.e. Ala1060 and Glu1061 have the same bond length. These results suggest that the overall protein structure is somehow altered by these different interactions with nearby residues, leading to functional defects in the protein.

## 4. Materials and Methods

### 4.1. Clinical Evaluation

We retrospectively enrolled 246 Caucasian patients diagnosed with lymphedema in hospitals across Italy. No consanguinity was reported in their families. Clinical diagnosis of lymphedema was made according to generally accepted criteria. The diagnosis of lymphedema was confirmed by three-phase lymphoscintigraphy according to the protocol of Bourgeois. Lymphoscintigraphy analysis was also performed for the mother of the proband of Family 1. Genetic testing was performed on germline DNA extracted from saliva or blood of the proband. Segregation analysis was performed on DNA extracted from the saliva of the proband’s relatives.

### 4.2. Genetic Analysis

A custom-made oligonucleotide probe library was designed to capture all coding exons and flanking exon/intron boundaries (±15 bp) of 29 genes known to be associated with lymphedema ^16^. We added the candidate gene *TIE1* to our panel (OMIM 600222).

DNA from probands was analyzed for genetic variants. Variants with likely clinical significance were confirmed by bidirectional Sanger sequencing on a CEQ8800 Sequencer (Beckman Coulter).

We developed a Next-Generation Sequencing (NGS) protocol for the screening of the most frequently mutated genes, namely *ADAMTS3* (OMIM 605011), *CELSR1* (OMIM 604523), *EPHB4* (OMIM 600011), *FAT4* (OMIM 612411), *FLT4* (OMIM 136352), *FOXC2* (OMIM 602402), *GATA2* (OMIM 137295), *GJA1* (OMIM 121014), *GJC2* (OMIM 608803), *HGF* (OMIM 142409), *KIF11* (OMIM 148760), *PIEZO1* (OMIM 611184), *PTPN14* (OMIM 603155), *SOX18* (OMIM 601618), and *VEGFC* (OMIM 601528), including the candidate gene *TIE1* (OMIM 600222).

We searched the international databases dbSNP and Human Gene Mutation Database professional (QIAGEN, CA, United States) for all nucleotide changes. In-silico evaluation of the pathogenicity of sequence changes in TIE1 was performed using the Variant Effect Predictor tool [22] and MutationTaster [23]. Minor allele frequencies were checked in the Genome Aggregation Database (gnomAD) [24]. All variants were evaluated according to American College of Medical Genetics and Genomics guidelines [25]. Detailed pre-test genetic counseling was provided to all subjects, who were then invited to sign informed consent to use of their anonymized genetic results for research.

### 4.3. In-Silico Analysis

The primary amino acid sequences of TIE1 in FASTA format (Table 4 and Table 5) were used as targets to search the Swiss model template library (SMTL) version 2019-10-24 and Protein Data Bank (PDB) release 2019-10-18 [26] for matching evolution-related structures by means of BLAST (Basic Local Alignment Search Tool) [27] and HHBlits [28]. Models were based on target-template alignment using ProMod3 of the SWISS-MODEL server [29]. Coordinates conserved between the target and the template were copied from the template to the model. Insertions and deletions were remodeled using a fragment library. Side chains were then rebuilt. Finally, the geometry of the resulting model was regularized with the CHARMM27 force field [30]. In the case of failure of loop modeling with ProMod3, an alternative model was built with ProMod-II [31]. Global and per-residue model quality were assessed using the QMEAN scoring function [32]. BioVia Discovery Studio Visualizer ver17.2 [33] was used to visualize the modeled protein, to vary the targeted amino acids and to analyze interactions at molecular level.

## 5. Conclusions

Despite the efforts and progress made in recent years, the genetics of lymphedema are still not entirely clear. Overall, most lymphedema patients remain without a genetic diagnosis since the genes currently associated with lymphedema account for about 25–30% of patients [16,17]. Understanding the genetics of lymphatic malformations may help to develop better therapies for lymphedema. In this report, we attempted to link data in the literature and NGS analysis of *TIE1* variants in lymphedema patients to investigate the role of the *TIE1* gene in the development of lymphatic system malformations and predisposition for lymphedema. Based on our results, we propose *TIE1* as a candidate gene for genetic testing of lymphedema. 

## Figures and Tables

**Figure 1 ijms-21-06780-f001:**
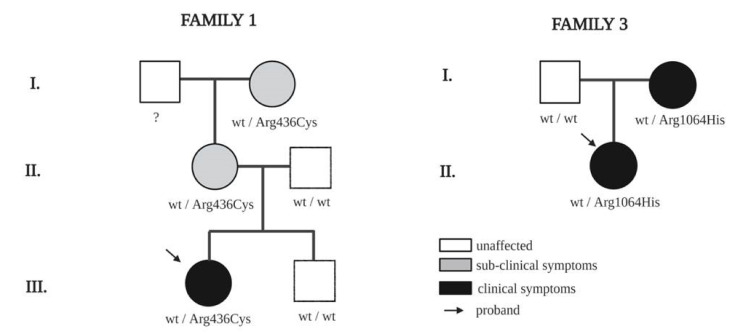
Pedigrees of families with *TIE1* variants through three familial generations (I, II and III) in family one and through two familial generations (I and II).

**Figure 2 ijms-21-06780-f002:**
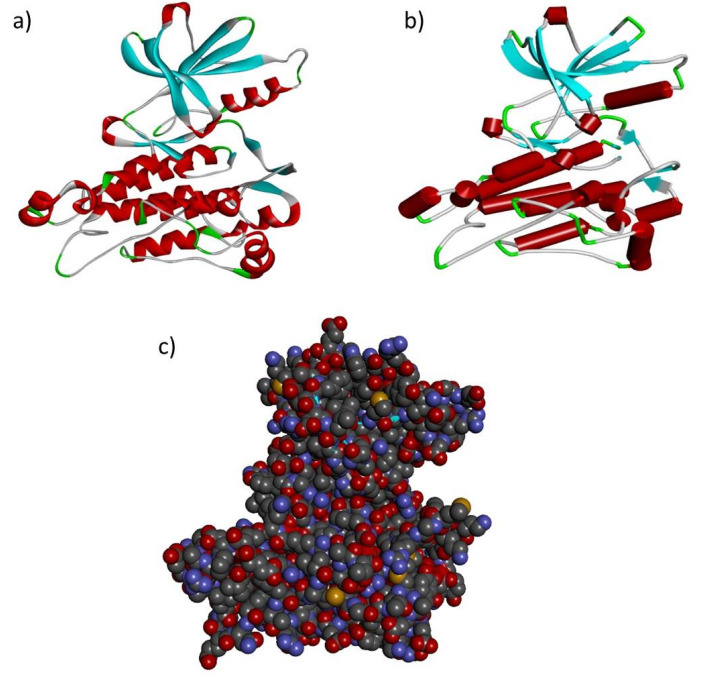
Modelled structure of the *TIE1* gene in (**a**) ribbon (**b**) schematic and (**c**) CPK view. Cyan regions indicate beta sheets, white indicates loops and red indicates alpha helices.

**Figure 3 ijms-21-06780-f003:**
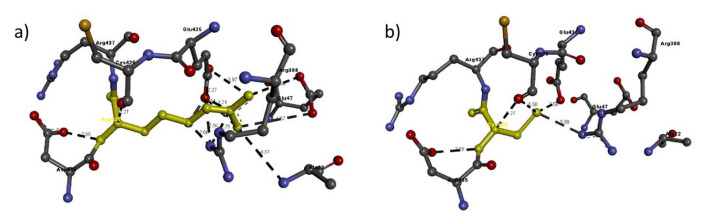
Molecular interactions of (**a**) Arg436 and (**b**) Cys436 (in yellow) of the modeled TIE1 protein with adjacent residues.

**Figure 4 ijms-21-06780-f004:**
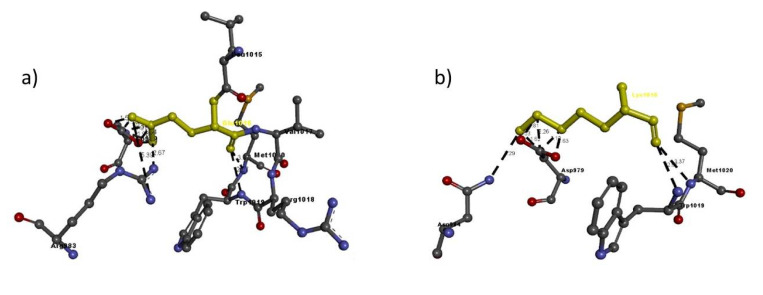
Molecular interactions of (**a**) Glu1016 and (**b**) Lys1016 (in yellow) of the modeled TIE1 protein with adjacent residues.

**Figure 5 ijms-21-06780-f005:**
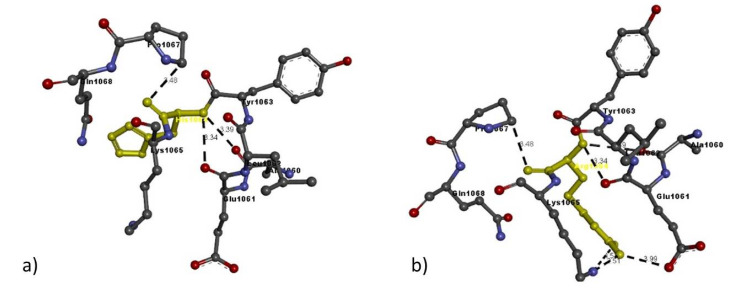
Molecular interactions of (**a**) Arg1064 and (**b**) His1064 (in yellow) of the modeled TIE1 protein with adjacent residues.

**Table 1 ijms-21-06780-t001:** Phenotype of *TIE1* mouse models.

Organism	Gene	Variant	Embryonic Lethality	Lymphatic Phenotype
Mouse	*TIE1*	*TIE1^−/−^*	E13.5 due to hemorrhage and cardiovascular malformations [5,6]	Edema at E12.5, abnormal patterning of lymphatic sacs and impaired integrity of lymphatic capillaries [7]
Mouse	*TIE1*	*TIE1^neo/neo^* hypomorphic allele	After E18.5, although some mice survived to adulthood	Edema, lymphatic vasculature abnormalities, overgrown jugular lymphatic vessels, dysfunctional fluid drainage in skin [8]
Mouse	*TIE1*	Deletion of intracellular domain of TIE1	E18.5; some mice were born alive, but none survived	Subcutaneous edema at E13.5, failed primary lymphatic system remodeling, lymphatic system malformation in newborn mice with induced mutation [10]
Mouse	*TIE1*	Conditional deletion in lymphatic endothelium	E18.5 (conditional mutation)	Lymphatic vessel remodeling failure, abnormal lymphatic valve morphogenesis, impaired maturation of the lymphatic system and failed development of lymphatic collecting vessels [9]

**Table 2 ijms-21-06780-t002:** Clinical features of probands with *TIE1* variants.

Family	Pedigree	Sex	Age	Clinical Features	Age of Onset	Familial	Variant Nomenclature
1	Proband	F	23	Edema of lower limbs	13	NO	NM_001253357.1:c.1306C>T/wt; NP_001240286.1:p.Arg436Cys
1	Father	M	52	Healthy	/	NO	wt/wt
1	Mother	F	49	Healthy	/	NO	NM_001253357.1:c.1306C>T/wt; NP_001240286.1:p.Arg436Cys
1	Brother	M	19	Healthy	/	NO	wt/wt
1	Grandmother	F	72	Healthy	/	NO	NM_001253357.1:c.1306C>T/wt; NP_001240286.1:p.Arg436Cys
2	Proband	F	52	Edema of right lower limb	25	NO	NM_001253357.1:c.3046G>A/wt NP_001240286.1:p.Glu1016Lys
3	Proband	F	47	Lymphedema of lower limbs, from the knee down	15	YES	NM_001253357.1:c.3191G>A/wt; NP_001240286.1:p.Arg1064His
3	Father	M	74	Healthy	/	YES	wt /wt
3	Mother	F	72	Lymphedema	/	YES	NM_001253357.1:c.3191G>A/wt; NP_001240286.1:p.Arg1064His

**Table 3 ijms-21-06780-t003:** Characterization of *TIE1* variants.

Variant	dbSNP ID	SIFT	PolyPhen	Frequency
TIE1:NM_001253357.1:c.1306C>T: NP_001240286.1:p.Arg436Cys	rs139244400	Deleterious	Possibly damaging	0.00002847
TIE1:NM_001253357.1:c.3046G>A:NP_001240286.1:p.Glu1016Lys	rs760492428	Deleterious	Probably damaging	0.0000325
TIE1:NM_001253357.1:c.3191G>A:NP_001240286.1:p.Arg1064His	rs34993202	Deleterious	Probably damaging	0.0007575

**Table 4 ijms-21-06780-t004:** Primary amino acid sequence used to search for templates in order to build models for TIE1.

**MVWRVPPFLLPILFLASHVGAAVDLTLLANLRLTDPQRFFLTCVSGEAGAGRGSDAWGPPLLLEKDDRIVRTPPGP** **PLRLARNGSHQVTLRGFSKPSDLVGVFSCVGGAGARRTRVIYVHNSPGAHLLPDKVTHTVNKGDTAVLSARVHK** **EKQTDVIWKSNGSYFYTLDWHEAQDGRFLLQLPNVQPPSSGIYSATYLEASPLGSAFFRLIVRGCGAGRWGPGCTK** **ECPGCLHGGVCHDHDGECVCPPGFTGTRCEQACREGRFGQSCQEQCPGISGCRGLTFCLPDPYGCSCGSGWRGS** **QCQEACAPGHFGADCRLQCQCQNGGTCDRFSGCVCPSGWHGVHCEKSDRIPQILNMASELEFNLETMPRINCAA** **AGNPFPVRGSIELRKPDGTVLLSTKAIVEPEKTTAEFEVPRLVLADSGFWECRVSTSGGQDSRRFKVNVKVPPVPLA** **APRLLTKQSRQLVVSPLVSFSGDGPISTVRLHYRPQDSTMDWSTIVVDPSENVTLMNLRPKTGYSVRVQLSRPGEG** **GEGAWGPPTLMTTDCPEPLLQPWLEGWHVEGTDRLRVSWSLPLVPGPLVGDGFLLRLWDGTRGQERRENVSSPQ** **ARTALLTGLTPGTHYQLDVQLYHCTLLGPASPPAHVLLPPSGPPAPRHLHAQALSDSEIQLTWKHPEALPGPISKYV** **VEVQVAGGAGDPLWIDVDRPEETSTIIRGLNASTRYLFRMRASIQGLGDWSNTVEESTLGNGLQAEGPVQESRAA** **EEGLDQQLILAVVGSVSATCLTILAALLTLVCIRRSCLHRRRTFTYQSGSGEETILQFSSGTLTLTRRPKLQPEPLSYPV** **LEWEDITFEDLIGEGNFGQVIRAMIKKDGLKMNAAIKMLKEYASENDHRDFAGELEVLCKLGHHPNIINLLGACK** **NRGYLYIAIEYAPYGNLLDFLRKSRVLETDPAFAREHGTASTLSSRQLLRFASDAANGMQYLSEKQFIHRDLAARNV** **LVGENLASKIADFGLSRGEEVYVKKTMGRLPVRWMAIESLNYSVYTTKSDVWSFGVLLWEIVSLGGTPYCGMTCA** **ELYEKLPQGYRMEQPRNCDDEVYELMRQCWRDRPYERPPFAQIALQLGRMLEARKAYVNMSLFENFTYAGIDATA** **EEA**

**Table 5 ijms-21-06780-t005:** Top ten models for 3D modelling of TIE1 structure.

Template	Seq Identity	Oligo-State	QSQE	Found by	Method	Resolution	Seq Similarity	Coverage	Description
4k0v.1.A	38.87	monomer	−	HHblits	X-ray	4.51 Å	0.40	0.45	TEK tyrosine kinase variant
4k0v.1.A	39.18	monomer	−	BLAST	X-ray	4.51 Å	0.40	0.43	TEK tyrosine kinase variant
2gy5.1.A	38.70	monomer	−	HHblits	X-ray	2.90 Å	0.40	0.37	Angiopoietin-1 receptor
2gy7.1.A	38.70	monomer	−	HHblits	X-ray	3.70 Å	0.40	0.37	Angiopoietin-1 receptor
2gy5.1.A	39.31	monomer	−	BLAST	X-ray	2.90 Å	0.41	0.36	Angiopoietin-1 receptor
2gy7.1.A	39.31	monomer	−	BLAST	X-ray	3.70 Å	0.41	0.36	Angiopoietin-1 receptor
1fvr.1.A	81.01	monomer	−	BLAST	X-ray	2.20 Å	0.56	0.28	Tyrosine-Protein Kinase TIE-2
6mwe.2.A	81.01	monomer	−	BLAST	X-ray	2.05 Å	0.56	0.28	Angiopoietin-1 receptor
2wqb.1.A	79.81	monomer	−	BLAST	X-ray	2.95 Å	0.55	0.28	Angiopoietin-1 Receptor
3I8p.1.A	81.01	monomer	−	BLAST	X-ray	2.40 Å	0.56	0.28	Angiopoietin-1 receptor

**Table 6 ijms-21-06780-t006:** Details of molecular interactions of Arg436 and Cys436 of the modeled TIE1 protein with adjacent residues.

Mutation	Amino Acid	Molecular Interactions	Bond Length in Å	Bond Type
Arg436Cys	Arg436	Arg388:N-Arg436:C	2.06	H-bond
		Arg388:N-Arg436:N	2.04	H-bond
		Arg388:N-Arg436:C	2.25	H-bond
		Glu425:C-Arg436:N	2.27	H-bond
		Glu425:O-Arg436:C	1.68	H-bond
		Glu425:O-Arg436:N	1.21	H-bond
		Glu425:O-Arg436:C	1.71	H-bond
		Arg436:N-Glu47:O	4.67	H-bond
		Arg436:N-Glu425:O	2.43	H-bond
		Arg436:N-Glu47:O	4.86	H-bond
		Arg436:N-Glu425:O	3.97	H-bond
		Arg436:N-Asp435:O	2.88	H-bond
		Arg436:C-Cys426:O	3.27	H-bond
		Ala22:N-Arg436:N	4.37	H-bond
	Cys436	Arg388:N-Cys436:S	3.09	H-bond
		Cys436:N-Asp435:O	2.88	H-bond
		Cys436:S-Glu425:O	2.50	H-bond
		Cys436:S-Cys426:O	3.30	H-bond
		Cys436:C-Cys426:O	3.27	H-bond

**Table 7 ijms-21-06780-t007:** Details of molecular interactions of Glu1016 and Lys1016 of the modeled TIE1 protein with adjacent residues.

Mutation	Amino Acid	Molecular Interactions	Bond Length in Å	Bond Type
Glu1016Lys	Glu1016	Asp979:C-Glu1016:C	2.10	H-bond
		Asp979:C-Glu1016:O	1.73	H-bond
		Asp979:C-Glu1016:O	1.98	H-bond
		Asp979:O-Glu1016:C	1.54	H-bond
		Asp979:O-Glu1016:O	1.91	H-bond
		Asp979:O-Glu1016:O	0.88	H-bond
		Asp979:O-Glu1016:O	1.56	H-bond
		Arg983:N-Glu1016:O	5.39	H-bond
		Arg983:N-Glu1016:O	2.67	H-bond
		Trp1019:N-Glu1016:O	2.97	H-bond
		Met1020:N-Glu1016:O	3.37	H-bond
	Lys1016	Asp979:C-Lys1016:C	2.12	H-bond
		Asp979:C-Lys1016:C	2.26	H-bond
		Asp979:C-Lys1016:N	1.62	H-bond
		Asp979:O-Lys1016:C	1.63	H-bond
		Asp979:O-Lys1016:C	1.81	H-bond
		Lys1016:N-Asp979:O	0.53	H-bond
		Trp1019:N-Lys1016:O	2.97	H-bond
		Met1020:N-Lys1016:O	3.37	H-bond
		Asn984:N-Lys1016:N	3.29	H-bond

**Table 8 ijms-21-06780-t008:** Details of molecular interactions of Arg1064 and His1064 of the modeled TIE1 protein with adjacent residues.

Mutation	Amino Acid	Molecular Interactions	Bond Length in Å	Bond Type
Arg1064His	Arg1064	Arg1064:N-Glu1061:O	3.99	H-bond
		Arg1064:N-Ala1060:O	3.39	H-bond
		Arg1064:N-Glu1061:O	3.34	H-bond
		Arg1064:N-Lys1065:N	4.57	H-bond
		Arg1064:N-Lys1065:N	2.51	H-bond
	Lys1016	His1064:N-Ala1060:O	3.39	H-bond
		His1064:N-Glu1061:O	3.34	H-bond
		Pro1067:C-His1064:O	3.48	H-bond

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
