# Peer review of "TIE1* as a Candidate Gene for Lymphatic Malformations with or without Lymphedema"

_ijms, 2020, doi:10.3390/ijms21186780_

Round 1
Reviewer 1 Report
The aim of the project was to identify different variants in the TIE1 gene that could be associated with lymphatic malformations or dysfunction and predisposition for lymphedema. The authors investigated 235 Italian lymphedema patients, who tested negative for variants in gene already associated with lymphedema. The data show that three probands carried different variants in the TIE1 gene. Two probands were related with lymphedema or lymphatic dysfunction in the cases of two families. The authors conclude that variants in TIE1 could contribute to the onset of lymphedema hence TIE1 could be a candidate gene for comprehensive genetic testing of lymphedema.
The paper is novel and interesting. I suggest some minor revision before the paper can be published.
Author Response
Dear Editor,
we thank the Reviewers for their helpful comments aimed at improving our manuscript. We are happy that they liked it and that its novelty was appreciated. We carefully considered all the points raised by the Reviewers and amended our manuscript accordingly.
Please find attached a point-by-point response to the Reviewers.
We hope that our manuscript will be now considered acceptable for publication in the International Journal of Molecular Sciences and look forward to hearing from you.
Best regards,
Dr Mirko Baglivo
On behalf of all authors
Reviewer 1
The aim of the project was to identify different variants in the TIE1 gene that could be associated with lymphatic malformations or dysfunction and predisposition for lymphedema. The authors investigated 235 Italian lymphedema patients, who tested negative for variants in gene already associated with lymphedema. The data show that three probands carried different variants in the TIE1 gene. Two probands were related with lymphedema or lymphatic dysfunction in the cases of two families. The authors conclude that variants in TIE1 could contribute to the onset of lymphedema hence TIE1 could be a candidate gene for comprehensive genetic testing of lymphedema.
The paper is novel and interesting. I suggest some minor revision before the paper can be published.
We are glad that this Reviewer found merit in our work.
Reviewer 2 Report
In the present study, the authors seek to identify Tie1 gene variants that associate with lymphatic malformations or dysfunction and predisposition for lymphedema. To do so, they examined a cohort of Italian lymphedema patients who did not express genes already known to associate with lymphedema. Tie1 was of particular interest as numerous murine genetic studies have demonstrated a role for Tie1 in lymphatic vasculogenesis and function, and they identified a 1.3% occurrence of Tie1 variants. Because two of the identified probands segregated with lymphedema or lymphatic dysfunction in familial cases, the authors propose the use of Tie1 variant to characterize lymphedema occurrence. The study is well-written and may advance understanding of lymphedema by identifying variants in Tie1, however, it is limited by its small sample size.
I have the following major comments/criticisms:
- Literature suggests the diagnosis and current genetic testing of lymphedema is complex and offers room for improvement. Please include more information on the current state and limitations of diagnosis and care in the introduction or discussion.
- Please measure the occurrence of these 3 Tie1 variants across a larger sample size. What is the frequency of these variants among the cohort that tested positive for variants in already known lymphedema genes? At what rate are they found in a healthy control population?
- Is it known which residues of TIE1 are involved in its interactions / complex formation with TIE2?
Author Response
Dear Editor,
we thank the Reviewers for their helpful comments aimed at improving our manuscript. We are happy that they liked it and that its novelty was appreciated. We carefully considered all the points raised by the Reviewers and amended our manuscript accordingly.
Please find attached a point-by-point response to the Reviewers.
We hope that our manuscript will be now considered acceptable for publication in the International Journal of Molecular Sciences and look forward to hearing from you.
Best regards,
Dr Mirko Baglivo
On behalf of all authors
Reviewer 2
In the present study, the authors seek to identify Tie1 gene variants that associate with lymphatic malformations or dysfunction and predisposition for lymphedema. To do so, they examined a cohort of Italian lymphedema patients who did not express genes already known to associate with lymphedema. Tie1 was of particular interest as numerous murine genetic studies have demonstrated a role for Tie1 in lymphatic vasculogenesis and function, and they identified a 1.3% occurrence of Tie1 variants. Because two of the identified probands segregated with lymphedema or lymphatic dysfunction in familial cases, the authors propose the use of Tie1 variant to characterize lymphedema occurrence. The study is well-written and may advance understanding of lymphedema by identifying variants in Tie1, however, it is limited by its small sample size.
We are glad that this Reviewer found merit in our work. We addressed the other comments as follows:
- Literature suggests the diagnosis and current genetic testing of lymphedema is complex and offers room for improvement. Please include more information on the current state and limitations of diagnosis and care in the introduction or discussion.
Thank you for the suggestion. Overall, genetic mutations can usually explain the etiology 25-30% of patients with lymphedema[1], [2]. Therefore, additional genes that are mutated in primary lymphedema are highly likely to exist. We included this information as requested in the introduction and conclusion.
- Please measure the occurrence of these 3 Tie1 variants across a larger sample size. What is the frequency of these variants among the cohort that tested positive for variants in already known lymphedema genes? At what rate are they found in a healthy control population?
We did screen the presence of these three variants in a series of other 50 patients that were enrolled in our laboratory, classified positive or that carried variant/variants in known genes that cannot clearly explain the lymphedema phenotype. In this cohort we did not found the presence of those variants, that seems to be rare. In addition, in a healthy control population, the gnomAD cohort, as indicated in the result section of the manuscript, we found that:
-the variant rs139244400 is found in 8 out of 251136 chromosome analyzed, in particular 0 individuals are homozygous for the allele with the variant (frequency of 0.0000319);
-the variant rs760492428 is found in 7 out of 143130 chromosomes analyzed in a healthy control population, in particular 0 individuals are homozygous for the variant (frequency of 0.0000489);
-the variant rs34993202 is found in 122 out of 157964 alleles examined, 0 homozygous individuals (frequency of 0.000772).
- Is it known which residues of TIE1 are involved in its interactions / complex formation with TIE2?
A paper of Veli-Matti Leppänen et coworkers on crystal structures of Tie1 and 2 receptors show that “the membrane-proximal Fn3 domains in Tie2 and Tie1 contribute to Tie2/Tie1 heterodimerization”. Their model also suggests an additional salt bridge between Tie1 Arg697and Tie2 Asp682. They state however that alternative interactions (i.e. electrostatic interactions) for Tie1 and Tie2 cannot be excluded[3].
[1] P. Brouillard, L. Boon, and M. Vikkula, “Genetics of lymphatic anomalies,” J. Clin. Invest., vol. 124, no. 3, pp. 898–904, 2014.
[2] P. E. Maltese et al., “Increasing evidence of hereditary lymphedema caused by CELSR1 loss-of-function variants.,” Am. J. Med. Genet. A, vol. 179, no. 9, pp. 1718–1724, Sep. 2019.
[3] V. M. Leppänen, P. Saharinen, and K. Alitalo, “Structural basis of Tie2 activation and Tie2/Tie1 heterodimerization,” Proc. Natl. Acad. Sci. U. S. A., vol. 114, no. 17, pp. 4376–4381, 2017.
Round 2
Reviewer 2 Report
The authors have adequately responded to all of my concerns.
Author Response
Dear Editor,
Thank you for your reminder. We carefully considered all the points raised by the Reviewer 2 and the Editor and have made the necessary corrections. We enclose our point-by-point response to the Reviewers.
We trust that our manuscript can now be accepted for publication in the International Journal of Molecular Sciences.
Yours sincerely,
Dr Mirko Baglivo
On behalf of all authors
Reviewer 2
The authors have adequately responded to all of my concerns. English language and style are fine/minor spell check required.
REPLY. Thank you. A native English-speaking scientific translator reviewed the final manuscript.